# Micro-Expression Recognition Using Uncertainty-Aware Magnification-Robust Networks

**DOI:** 10.3390/e24091271

**Published:** 2022-09-09

**Authors:** Mengting Wei, Yuan Zong, Xingxun Jiang, Cheng Lu, Jiateng Liu

**Affiliations:** 1Key Laboratory of Child Development and Learning Science of Ministry of Education, Southeast University, Nanjing 210096, China; 2School of Biological Science and Medicial Engineering, Southeast University, Nanjing 210096, China; 3School of Information Science and Engineering, Southeast University, Nanjing 210096, China

**Keywords:** micro-expression recognition, micro-expression magnification, locality sensitive hashing, uncertainty, self-attention

## Abstract

A micro-expression (ME) is a kind of involuntary facial expressions, which commonly occurs with subtle intensity. The accurately recognition ME, a. k. a. micro-expression recognition (MER), has a number of potential applications, e.g., interrogation and clinical diagnosis. Therefore, the subject has received a high level of attention among researchers in affective computing and pattern recognition communities. In this paper, we proposed a straightforward and effective deep learning method called uncertainty-aware magnification-robust networks (UAMRN) for MER, which attempts to address two key issues in MER including the low intensity of ME and imbalance of ME samples. Specifically, to better distinguish subtle ME movements, we reconstructed a new sequence by magnifying the ME intensity. Furthermore, a sparse self-attention (SSA) block was implemented which rectifies the standard self-attention with locality sensitive hashing (LSH), resulting in the suppression of artefacts generated during magnification. On the other hand, for the class imbalance problem, we guided the network optimization based on the confidence about the estimation, through which the samples from rare classes were allotted greater uncertainty and thus trained more carefully. We conducted the experiments on three public ME databases, i.e., CASME II, SAMM and SMIC-HS, the results of which demonstrate improvement compared to recent state-of-the-art MER methods.

## 1. Introduction

In recent years, micro-expressions (MEs) have gained increasing publicity in both the academic and industrial community. A micro-expression is a stifled facial expression with subtle and spontaneous muscle movements that appears very briefly (i.e., less than 200 ms). It usually appears when people attempt to conceal their true emotional states [1] and is hard to disguise even for expert actors. Therefore, MEs can be regarded as reliable clues with which to infer human emotions, making them especially helpful in high-stake situations, e.g., in the judicial system, police interrogation [2] and clinical diagnosis. However, revealing spatial-temporal information is difficult because of the low intensity [3] and short duration of MEs and it is rather challenging for humans to identify a ME with the naked eye. In order to improve Micro-Expression Recognition (MER) performance, psychological researchers have made contributions so as to train people to use training tools. Despite these tools, the ability of people to recognize MEs achieves a less than 40% accuracy. Moreover, manually recognizing MEs is time consuming, urging people to seek more automatic and accurate methods for MER [2,4,5,6].

Facial expressions present in a dynamic manner, MEs, can be recorded in online and offline videos. This characteristic facilitates the creation of ME databases, which provide the ability to perform insightful studies on MER. The USF-HD [7], Polikovsky’s Database [3] and York Deception Detection Test (York-DDT) [8] are the earliest databases collecting ME video clips. However, these databases are not widely used because of their limitations. For instance, the MEs in Polikovsky’s Database and USF-HD are collected by asking participants to pose intentionally instead of poses being elicited unconsciously. In addition, the sample size is insufficient in the York-DDT, which limits the more intensive implementation of an ME analysis. Considering the spontaneity of eliciting MEs, state-of-the-art databases collect ME samples induced in the laboratory, e.g., SMIC [9], CASME [10], CASME II [11], ${\mathrm{CAS}(\mathrm{ME})}^{2}$ [12] and SAMM [13], which is a great improvement compared with the previous databases. Based on these datasets, many approaches have been devised to promote the development of MER systems. In preliminary studies, researchers usually use low-level features, e.g., Local Binary Pattern (LBP) [14], optical flow [15], gradient-based features [3] and their variants to describe ME images. These features provide a form of visual clues to be extracted from the details of the image, such as intensity which changes either temporally or as a gradient. However, most of them lack the explicit semantic interpretation of ME itself and are overdependent on hand gestures. Therefore, recent studies focused on high-level features utilizing deep learning.

A high-level feature is the combination of multiple low-level features. It is semantically interpretable and has a more discriminative ability to represent MEs. Most of the-state-of-the-art high-level representations are extracted from CNN models. In some earlier studies [16,17,18,19], researchers mainly focused on spatial information, some of which provide clues based on ME frames and the others based on optical flow. Recent works attempted to encode both spatial and temporal information for a more comprehensive representation since facial movement variation is a dynamic process. Many deep-learning-based methods specialize in capturing dependencies in long sequences, e.g., 3D Convolutional Neural Networks (3D CNNs) [20,21], and Long Short-Term Memory Networks (LSTMs) [22], which are employed to capture the motion of MEs. Nevertheless, MEs are reflected in the local area with a low intensity of muscle movements, resulting in the perception of motion variation being complex. To tackle this issue, a technique to magnify these subtle movements can be helpful to improve the performance of MER. The most commonly used magnification techniques, e.g., Eulerian Motion Magnification (EMM) [23], Global Lagrangian Motion Magnification (GLMM) [24], and Learning-based Video Motion Magnification (LVMM) [25], have shown good performance in magnifying subtle movements. Inspired by this effect, many MER works introduced magnification techniques to magnifying ME intensity, and their results proved the effectiveness of ME magnification [23,24,26,27].

Despite these achievements, the following problem that occurs during magnification is rarely mentioned: the unified-magnification strategy, which is implemented by setting the same amplification factor for all ME samples, is not adaptable, since the intensity variation scope varies in different ME samples. For example, in some ME video clips, the intensity variation is more acute, so a small amplification factor may be adequate to magnify them. Conversely, in other clips, the variation is very subtle and requires larger amplification factors. An improper magnification level may introduce artefacts and even cause severe deformation, as shown in Figure 1. Therefore, a method to effectively capture powerful motion representation while avoiding the deficiencies of magnification is required. Moreover, existing methods are limited due to imbalanced ME databases. For the minority classes, the model sees much fewer samples and thus tends to underfit these samples during training. This problem can impair the interpretation of unfamiliar ME samples.

In this paper, we proposed a novel framework, namely an Uncertainty-Aware Magnification-Robust Network (UAMRN), to address the aforementioned problems. Specifically, in order to make the network perceive ME movements more easily, we used magnification techniques to rebuild a sequence with a more discriminative representation for motion. Afterwards, we imposed sparsity constraints via Locality Sensitive Hashing (LSH) into a self-attention block, adaptively reserving highly-correlated ME features and discarding the uncorrelated ones. The resultant sparsity of the self-attention feature retains the global modeling ability of the standard self-attention feature while mitigating magnification artefacts. On the other hand, to manage the imbalanced dataset issue, we utilized sample uncertainty information based on the Bayesian uncertainty measure to learn more generalizable features for rare classes.

This work is an extended version of our conference paper (part of this work is presented in ICPR2022). In addition, a more robust model with higher MER performance, enhanced by using uncertainty estimation to guide the training, is provided. The experiments were reimplemented to test the model, and the results are provided. The main contributions are summarized as follows:A sequence more able to reflect ME movements was rebuilt via magnification techniques.To preserve as many of the ME-related features as possible and suppress magnification noise, we proposed a sparse self-attention (SSA) block by enforcing sparsity in attention terms using Locality Sensitive Hashing (LSH).To improve the recognition accuracy of rare classes, we estimated the uncertainty for each sample, where the samples from rare classes can be trained more carefully.Extensive experiments conducted on three widely used databases demonstrate that our approach yields competitive results compared with state-of-the-art MER methods.

## 2. Related Works

### 2.1. Motion Magnification for Micro-Expressions

It was previously shown on a wide spectrum of MER tasks that magnifying ME intensity before recognition can promote performance [23,28]. Generally, most MER works using developed magnification techniques follow the method of first magnifying original ME images and then sending them to the network for recognition. To ensure the classifier better distinguishes between different ME classes, Li et al. [26] adopted Eulerian Motion Magnification (EMM) to magnify the subtle motions in videos. Ngo et al. [24] then introduced a new magnification technique, i.e., Global Lagrangian Motion Magnification (GLMM), which was found to provide more discriminative magnification. However, these earlier MER works used hand-crafted filters to magnify ME intensity, which required more hyper-parameters settings and were prone to noise and blur. Therefore, as  deep learning methods emerge, recent works resorted to more robust magnification techniques, and their results demonstrated better flexibility. Lei et al. [25] adopted a transfer learning strategy to magnify MEs for better structural representation. Their later work used CNN-based methods for ME magnification and yielded further improvements [29]. Although learning-based magnification approaches are less condition limited, they have the same problem as hand-crafted filters when used to magnify MEs, i.e., the deficiency of adaptability to different subjects and artefacts introduced by improper amplification factors. Therefore, in our work, we devised a new role for magnification, implemented by setting multiple magnification levels for the same subject. Meanwhile, the artefacts of magnification can be mitigated via our sparsity design in self-attention blocks.

### 2.2. Uncertainty Estimation in Neural Networks

Uncertainty knowledge provides an estimation of the confidence of the Neural Network’s (NN’s) prediction. Hitherto, significant studies have been conducted in the computer vision domain to provide uncertainty estimates. In general, there are two forms of uncertainty which are as follows: [30] (1) model uncertainty resulting from an underfit model (epistemic uncertainty) and (2) data uncertainty due to the noise inherent in databases (aleatoric uncertainty), where model uncertainty can be decreased by adding more data while data uncertainty is only related to the observation itself. In earlier works, these uncertainties were evaluated using Bayesian deep methods [31], where the model weights were considered as a distribution instead of as fixed values. However, Bayesian inference comes with huge computational cost in deep models, so another simple technique, i.e., Monte-Carlo Dropout sampling [32] by placing a Bernoulli distribution over weights, is much more widely adopted. It has been shown that uncertainty to guide training could facilitate the ability to recognize samples contaminated with noise, especially for datasets of a small size [33,34,35]. Inspired by these works, we proposed an improvement of generalization ability based on uncertainty estimation, where the classes not fully learned by the model due to the lack of data, compared with those classes with more samples, are weighted higher during training.

## 3. Proposed Method

The framework of our proposed model is shown in Figure 2. First, we rebuilt a new sequence by enhancing ME intensity, where each image corresponded to a specific amplification factor (AF). The rebuilt sequence is much easier to recognize because it holds more distinct intensity variation compared with the original ME video clip. Afterwards, to mitigate improper magnification levels, we adaptively zeroed out some useless clues by imposing sparsity in the self-attention (SSA) block. Moreover, to handle the imbalanced ME dataset problem, we estimated the uncertainty score for each sample and used the score to guide the network to provide better training for those samples from rare classes.

### 3.1. ME Sequence Reconstruction

In the original ME video clip, the intensity variation was not clear enough to be captured due to the subtle movement of the ME. Therefore, we constructed a new sequence which presents a larger intensity scope, implemented by simulating the intensity variation with magnification techniques, as demonstrated in (a) of Figure 2. The new sequence discards the temporal meaning of the original where the intensity varies with time, but with amplification factors.

Using hand-crafted filters requires more human intervention, so we introduced a new magnification technique based on a transfer learning strategy, i.e., a deep network pre-trained on a large-scale database devised by Oh et al. [36], to magnify ME intensity. This strategy allows the network to function in an end-to-end manner and produced clearer magnified images. According to the definition of motion magnification provided by Wadhwa et al. [37], for a frame I(P,t) in a video at position P=(x,y), denoted as I(P,t)=f(P+σ(P,t)), where σ(P,t) represents the motion field at *P* and time *t*, the magnified image is as follows:(1)$$\tilde{I}=f\left(P+\left(1+\alpha \right)\sigma \left(P,t\right)\right),$$
where α is amplification factor. Here, in an ME video clip, we used the onset $I_{onset}$ and the apex $I_{apex}$ frame to produce magnified images, denoted as
(2)$$I_{mag}=f(I_{onset}+\left(1+\alpha \right)|I_{apex}-I_{onset}\left|\right),$$
where |·| represents the pixel-wise subtraction and $I_{mag}$ represents the magnified image. These magnified images are then arranged corresponding to their amplification factors (AFs) into a new sequence. Compared with the original video, the new one better reflects intensity variation since it is built on the magnified intensity discrepancy between frames.

### 3.2. Magnification-Robustness from Sparse Self-Attention

The new sequence was built based on the same set of AFs, which led to a prominent problem, i.e.,  uncontrollable artefacts resulted from improper magnification, especially from overlarge AFs. Therefore, some feature vectors from the sequence may be useless as they originate from magnified images with excessive deformation. To alleviate this issue, we removed these features with three sparse self-attention (SSA) blocks, where the sparsity was enforced in the standard self-attention feature.

#### 3.2.1. Self-Attention

According to [38], the standard self-attention block captures the dependencies in a sequence by enumerating all locations, mainly functioning using three learnable matrices, i.e., $Q\in R^{L\times d_{q}}$ to match others, $K\in R^{L\times d_{k}}$ to be matched and $V\in R^{L\times d_{v}}$ for the information to be extracted, which is formulated as
(3)$$\mathrm{Attention}\left(Q,K,V\right)=\mathrm{Softmax}\left(\frac{{\mathrm{QK}}^{T}}{\sqrt{d_{k}}}\right)V,$$
where $d_{q}=d_{k}$ and $d_{q},d_{k},d_{v}$ are the dimensions of query (q), key (k) and value (v), respectively. *L* represents the sequence length. In Equation (Equation 3), each $q_{i}$ is a dot product with each $k_{j}$ at every location of the sequence, where i,j∈{1,...,L}. Here, we expect the features from the magnified images with excessive deformation to be abandoned, and that the attention is operated only for the locations with highly-correlated features. Therefore, for some queries in Q, if they originate from the magnified frames with over-large deformation, there is likely no key in K that shares a high correlation with them. These cases can be set to zero in the term $\mathrm{Softmax}(\frac{{\mathrm{QK}}^{T}}{\sqrt{d_{k}}})$.

#### 3.2.2. Magnification-Robust Sparsity

Since we wanted to discard the less relevant elements, an instinctive way is to sort them according to their correlation and keep those larger features. However, the process of selecting the threshold cannot be adapted to recognize useless features, because each subject may correspond to different thresholds. Therefore, we devised the Sparse Self-Attention (SSA) block to make this process adaptive, as shown in (b) of Figure 2. More specifically, to ensure that the more relevant elements were more likely to be reserved in the attention matrix, we embedded Locality Sensitive Hashing (LSH) into the standard self-attention. The way that LSH operates is to take the space and cut it into several separate regions with hyper-planes. Each region represents a hash bucket. A vector is projected to the space by providing a dot product with the hyper-planes. If two vectors have a higher correlation, they are likely to fall into the same hash bucket. The resultant attention functions by only attending to the vectors in the same hash bucket. In the reconstructed sequence, the feature vectors containing the majority of the magnification artefacts fall into different hash buckets, as demonstrated in Figure 3. By detracting the attention from improperly magnified images and maintaining attention to highly-correlated ME features, our model can filter more discriminative clues to recognize MEs.

We provide an example to clarify this. Suppose there are three hyper-planes in the space, denoted as $h_{1}$, $h_{2}$ and $h_{3}\in R^{d}$, which cut the space into $2^{3}=8$ regions. Each region represents a hash bucket for holding vectors. For a feature vector $x\in R^{d}$, if the product $x^{T}h>0$, it is projected on one side of h with the tag “0”; otherwise it is labeled as “1”. For instance, if we compute $x^{T}h_{1}<0,x^{T}h_{2}<0$ and $x^{T}h_{3}>0$, the vector x will fall into the bucket tagged as h(x)=“001". Then, each bucket corresponds to a tag consisting of three binary digits. For brevity, the buckets are collected into a set denoted as H={“001",“000",“010",“011",“101",“100",“111",“110"}. To form the attention matrix ${\mathrm{QK}}^{T}$, we first hash all the queries and keys into the space and then only keep those in the same bucket. The resultant attention is implemented by attending to the queries and keys in the following index set:(4)$$\begin{array}{cc} \Omega_{i,j}= & \{\left(i,j\right)|h\left(q_{i}\right)=h\left(k_{j}\right)\},\\ & s.t.\;0<i,j<L. \end{array}$$

Considering that the hyper-planes are randomly generated, some highly-correlated elements may be hashed into different buckets. To mitigate this issue, we first generated multiple sets of hyper-planes $\left\{H^{\left(1\right)},H^{\left(2\right)},...,H^{\left(n_{t}\right)}\right\}$, and then each set was hashed once by the queries and keys. The result is obtained by gathering the union of these sets which is formulated as follows:(5)$$\begin{array}{c} \Omega_{i,j}=\overset{n_{t}}{\underset{r=1}{⋃}}\Omega_{i,j}^{\left(r\right)}, \end{array}$$
where $\Omega_{i,j}^{\left(r\right)}=\left\{\left(i,j\right)|h^{\left(r\right)}\left(q_{i}\right)=h^{\left(r\right)}\left(k_{j}\right)\right\}$ and $n_{t}$ is the number of times we regenerate hyper-planes. Here, the original attention matrix in Formula (Equation 3) is rewritten as
(6)$$\mathrm{Attention}\left(Q,K,V\right)=\sum_{\left(i,j\right)\in \Omega_{i,j}}\mathrm{Softmax}\left(\frac{q_{i}k_{j}^{T}}{\sqrt{d_{k}}}\right)V.$$
Compared with Equation (Equation 3), the sparse one can retain the ability to globally attend to the most informative locations while avoiding the interference from improper magnification levels, achieved in an adaptive way. We provide the pseudocode in Algorithm 1.
**Algorithm 1:** Sparse Self-Attention with Locality Sensitive Hashing.
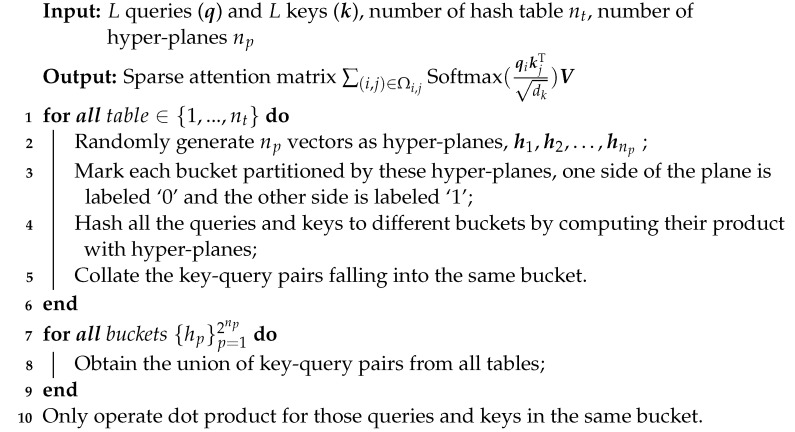


### 3.3. Uncertainty-Aware Training for the Imbalanced Dataset Problem

The imbalanced dataset problem, prevailing in ME databases, tends to degrade the generalization of rare classes of the dataset. Moreover, during the collection of ME samples, the expression presented by the participant may deviate from the initial intention that the stimulus materials aim to elicit. For instance, the stimulus material is a clip of video with disgusting contents intending to arouse people’s feeling of “disgust”, but the expression they demonstrate may be “fear”. The sample is still labeled as “disgust”. This mistake is inevitable in existing ME databases. Therefore, to handle label noise on imbalanced datasets, we estimated the epistemic and aleatoric uncertainty for each sample, and used them as the weights of the sample’s contribution to the loss. As the uncertainty derives from the uncertainty of the model weights, in our network, we added dropout after some layers in the SSA blocks. The output includes both predictive mean and its variance as follows:(7)$$\begin{array}{cc} \bar{y},{\hat{\sigma }}^{2}t & =f^{W}\left(x\right),\\ & p=\mathrm{Softmax}(\bar{y}), \end{array}$$
where f is our network parametrized by *W* and p is the class probability with $p_{c}$ as the element denoting that of class *c*. For a sample x, its prediction is subject to Gaussian distribution $\hat{y}∼N\left(\bar{y},{\hat{\sigma }}^{2}\right)$. The uncertainty for this sample is computed as the expectation of $\hat{y}$, approximated using Monte Carlo integration:(8)$$\gamma \approx \frac{1}{M}\sum_{m=1}^{M}{\hat{y}}_{m}^{2}-{\left(\frac{1}{M}\sum_{m=1}^{M}{\hat{y}}_{m}\right)}^{2}+\frac{1}{M}\sum_{m=1}^{M}{\hat{\sigma }}_{m}^{2},$$
where ${\left\{{\hat{y}}_{m},{\hat{\sigma }}_{m}^{2}\right\}}_{m=1}^{M}$ is the set of *M* sampled outputs. Here, we used the uncertainty as the weight of the sample contributing to the loss, so the objective becomes
(9)$$L=-\frac{1}{N}\sum_{n}\gamma_{n}\sum_{c=1}^{C}y_{n,c}\log p_{n,c}$$
where *C* is the number of classes, *n* is the index of sample, and *N* is the number of samples. During the training, as the samples with larger uncertainty mostly resulted from noisy labels or rare classes and contribute more to the loss, the model can learn to pay more attention to these samples in order to attenuate the loss. Therefore, these samples can be trained more carefully, leading to a more relaxed generalization boundary on rare classes.

## 4. Experiments

### 4.1. Databases

We conducted experiments on three public ME datasets, i.e., CASME II [11], SAMM [13], and SMIC-HS [9]. More details concerning these datasets are provided below.

**CASME II** contains 246 ME video clips elicited from 26 participants, recorded by a high-speed camera at 200 fps. The following five ME categories were selected for experimentation: happiness (32), disgust (63), repression (27), surprise (25) and other (99), where the number in the brackets denotes the number of samples in this class.

**SAMM** includes 159 ME samples from 32 participants of 13 ethnicities. The recording rate of the camera is 200 fps. We also selected five categories from this database, i.e., anger (57), happiness (26), contempt (12), surprise (15) and other (26).

**SMIC-HS** is the HS version of SMIC where the videos are recorded by a 100 fps high-speed camera. It collects 164 spontaneous MEs which are divided into the following classes: positive (51), negative (70), and surprise (43). We use all the samples in this dataset.

When we rebuild the new sequence using magnification, we need to know the indices of the apex and onset frame. The CASME II and SAMM provide the annotation of key frames while the SMIC-HS does not. Considering that our model does not require the accurate location of key frames as we can produce high intensity variation by applying a large amplification factor, we selected the middle frame as the apex.

### 4.2. Protocols

To evaluate the efficacy of the proposed method, we used the leave-one-subject-out (LOSO) cross-validation protocol, implemented by treating all samples from one subject as the test set while the remaining samples comprised the training set. The final recognition result is obtained by averaging all the subjects. The metrics of evaluation are the accuracy and F1-score, given as below:(10)$$\begin{array}{cc} \; & acc=\frac{T}{N},\\ F1-\mathrm{score} & =\frac{1}{c}\sum_{c=1}^{C}\frac{2\times P_{c}\times R_{c}}{P_{c}+R_{c}}, \end{array}$$
where the accuracy assesses the overall recognition performance, with *T* denoting the total number of correct predictions. F1-score evaluates the ability when managing the unbalanced database problem, where $P_{c}$ and $R_{c}$ are the precision and recall of the *c*-th micro-expression, respectively.

### 4.3. Implementation Details

In each ME video clip, we first used [39] to compute 68 facial landmarks of the onset, and then cropped the face area by these coordinates. The apex was cropped using the coordinates of the onset, aiming to ensure the two frames had no displacements caused by irrelevant factors. We resized all the images to 224×224 as the input.

Considering that a small ME dataset may result in overfitting, we used the first few layers of the pre-trained Resnet-18 [40], pre-trained on the FER+ [41] dataset, to extract shallow features of the magnified frames. These features were set as the input of the proposed framework. To encode position clues before SSA blocks, we adopted sine and cosine functions of different frequencies as follows: $PE_{\left(pos,2d\right)}=\sin\left(pos/10000^{2d/d_{model}}\right)$, $PE_{\left(pos,2d+1\right)}=\cos\left(pos/10000^{2d/d_{model}}\right)$, where pos is the position in the sequence and *d* is the specific dimension of $d_{model}$.

We trained the UAMRN at 40 epochs. For learning parameters, the decay rates for the 1st and 2nd moment estimates were 0.9 and 0.999, respectively, in the ADAM optimizer. The learning rate was set to $2\times 10^{-4}$ and reduced by 0.1 after every 10 epoch. Other detailed hyper-parameters are listed in Table 1.

### 4.4. Experimental Results

In Table 2, Table 3 and Table 4, we compare the performance of the UAMRN against other relelated MER works. The comparison is provided from two perspectives, i.e., a comparison against the methods using hand-crafted or learning based ME features, and against those methods with the aid of magnification techniques.

#### 4.4.1. Comparison with Methods Using Hand-Crafted Features and Deep Models

It can be observed that when compared with other traditional and deep features, our method obtained the best overall performance. To be specific, UAMRN exceeded early hand-crafted works, e.g., LBP-SIP [48], LBP-TOP + AdaBoost [42], and STRBP [43], by a large margin, which proves that deep models have more advantages in extracting ME features. Moreover, compared with most state-of-the-art learning based methods, e.g., TSCNN [44], 3D-CNNs [21], MERSiamC3D [20], UAMRN also achieves better results. TSCNN surpasses our method in terms of F1-score by 4.68% and of accuracy by 0.4%, respectively, on the CASME II. According to the ablation experiments reported, in its model, the dynamic-temporal and static-spatial modules are two major parts contributing to recognizing MEs. Our method also focused on temporal and spatial clues, where the temporal clue was enhanced by magnifying intensity. Therefore, the performance of our model on the other two databases is much higher.

#### 4.4.2. Comparison with Methods Magnifying ME Intensity

The proposed UAMRN obtained increases of 9.72% and 13.36% in the accuracy on the CASME II, compared with ME-Booster [23] and HIGO + Mag [26] which adopt Eulerian Motion Magnification (EMM) for magnification. Compared with the approaches using the same magnification techniques as ours, i.e., Graph-tcn [25], and AU-GCN [29], UAMRN also performs better. We speculate that its superiority lies in the innovative exertion of magnification, implemented by applying multiple AFs for the same subject instead of applying just one AF, which adds adaptability to the framework. In addition, it can be noticed that UAMRN exceeds most MER methods substantially in terms of the F1-score, indicating that our method has a better ability to manage the imbalanced dataset problem.

### 4.5. The Effectiveness of Locality Sensitive Hashing

In order to validate that the magnified images with excessive artefacts can be discarded through SSA blocks, we retrieved the feature vectors q and k and plotted the result $\sum_{\left(i,j\right)\in \Omega_{i,j}}\mathrm{Softmax}\left(\frac{q_{i}k_{j}^{T}}{\sqrt{d_{k}}}\right)V$. Afterwards, we verified them with the frames from the rebuilt sequence. Large indexes of q and k denote features from magnified frames with large AFs, as shown in Figure 4.

It can be observed that generally, non-zero elements are centred near the diagonal of the matrix, suggesting that adjacent frames tend to have a higher correlation than distant frames. Additionally, the majority of the lower right corner of the matrix, where the indexes of q and k are larger (corresponding to images with large AF), is filled with zero values, which validates the efficacy of LSH for removing distorted images. It can also be noted that some vectors sharing adjacent indexes were placed into different buckets, e.g., $k_{20},q_{9},k_{17}$. We speculate the reason is that some vectors, even if they are close to one another, may fall into different buckets due to the randomness of the way that hyper-planes are partitioned. The impact of this problem can be counteracted by employing attention with multi-heads to extract multiple levels of information.

## 5. Ablation Study

### 5.1. Uncertainty-Aware versus Deterministic

To evaluate the efficacy of uncertainty estimates, we also conducted experiments for comparison, which did not require forward propagation multiple times to compute uncertainty. The predictive mean $\bar{y}$ of the output is used as the prediction, so the optimization objective becomes the standard cross-entropy loss. The results are shown in Table 5.

It can be found in the table above that using uncertainty as the estimation to guide the training could improve the overall performance, especially in terms of the F1-score. During training, the formulation of epistemic and aleatoric uncertainty introduced a new loss function, whereby the sample that the network is less confident with in terms of prediction contributes more to the loss. With the learning of loss attenuation, the effect from minority classes and corrupted labels are also attenuated, leading to a more robust network for the data.

### 5.2. Full Attention versus Sparse Attention

In order to validate the performance of sparse self-attention in mitigating useless magnification clues, we conducted experiments by replacing the SSA blocks with standard full-attention. In addition to comparing the results in terms of accuracy and F1-score, we also conducted an analysis of their computational efficiency. In the SSA, the number of hash buckets ($n_{p}$) was exponential to that of the hyper-planes, as follows: $w=2^{n_{p}}$. Hence, a hash bucket has an averaged size $\frac{L}{n_{p}}$. For briefness, we only computed hashing once in the SSA, so the maximum time complexity is $w{\left(\frac{L}{2n_{p}}\right)}^{2}$. As for full-attention, the time complexity is quadratic to the length of the sequence, denoted as $bn_{h}L^{2}$. More results are shown in Table 6.

As shown in Table 6, the SSA performed better on the three databases than the standard one, which proved that imposing sparsity constraints could help the model retain the ability to capture the dependencies in sequences, while erasing defective magnification clues. The less relevant vectors can be allocated lower scores in the attention matrix without being set as zero. This case will not impact the result much when the sequence is clear. However, once the sequence is dominated with noise, the performance will decline critically since the attention matrix keeps all the products from the vectors. From the model’s perspective, a sequence filled with useless clues is identical to useful clues, because it implements attention without distinguishing locations. However, in the SSA, as the noise dominates the sequence, the attention matrix will be more sparse. Hence, the model can adaptively attend to those highly-correlated locations, indicating that knowing where to attend is more important than attending all.

## 6. Conclusions

In this paper, a novel uncertainty-aware magnification-robust network (UAMRN) was proposed to resolve the low intensity and imbalanced dataset problem in MER. To make ME movements more distinguishable, a surrogate sequence was built via a magnification technique. Subsequently, to mitigate the artefacts in the new sequence produced by magnification, and to adaptively implement useful magnification levels for different ME samples, we devised a sparse self-attention (SSA) block by incorporating sparsity into the standard self-attention feature using locality sensitive hashing. SSA adaptively attends to those highly-correlated locations and abandons the vectors corrupted with noise, leading to a more robust operation. Moreover, to manage the imbalanced dataset problem, we computed the uncertainty scores corresponding to each sample, where the scores were used as the weight to guide the training of the model. In this way, the samples from minority classes can be trained more carefully; hence, the features extracted from these classes were more generalizable. Extensive experiments on the three public ME databases helped to validate the efficacy of the proposed method.

## Figures and Tables

**Figure 1 entropy-24-01271-f001:**
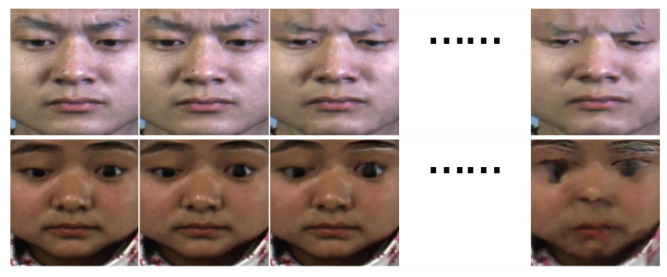
Two ME samples magnified with the same set of amplification factors. The first and second row display “disgust” and “surprise”, respectively. A large magnification level can facilitate the extraction of discriminative features for one sample, i.e., “disgust”, but can cause severe deformation on the other, i.e., “surprise”.

**Figure 2 entropy-24-01271-f002:**
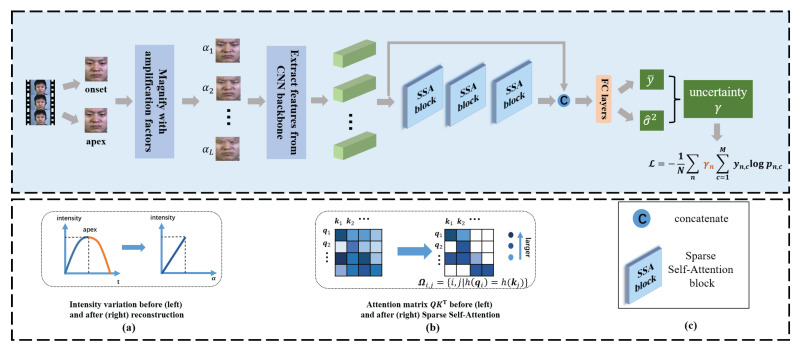
Framework of the proposed method. (**a**) denotes the intensity variation before and after sequence reconstruction. (**b**) denotes the attention matrix before and after enforcing sparsity. (**c**) is the interpretation of some signs in the above framework.

**Figure 3 entropy-24-01271-f003:**
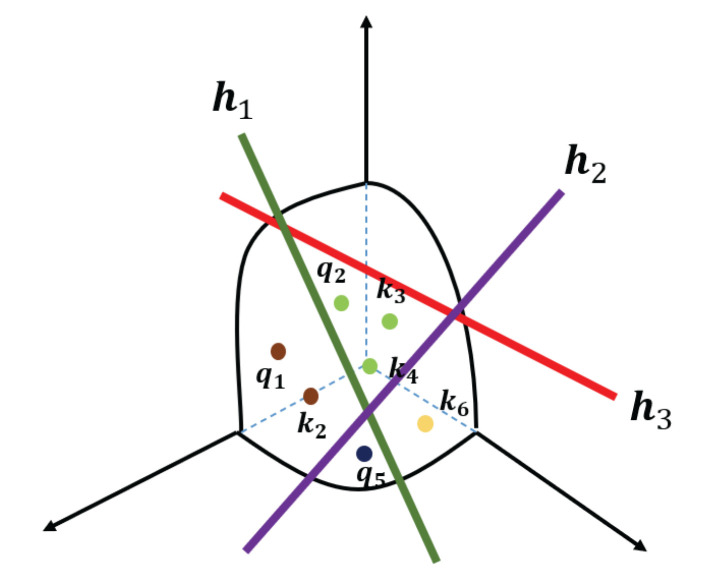
An example of locality sensitive hashing (LSH). Three hyper-planes $h_{1},h_{2},h_{3}$ cut the space into eight buckets where nearby feature vectors are more likely to be projected into the same bucket, e.g., ($q_{1},k_{2}$), ($q_{2},k_{3},k_{4}$). Vectors from magnification noise are hashed into other buckets discretely, e.g., $q_{5},k_{6}$.

**Figure 4 entropy-24-01271-f004:**
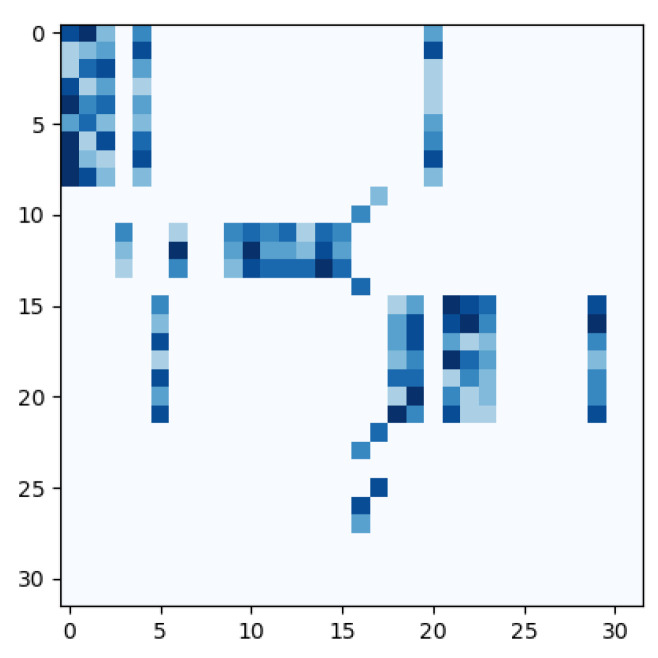
An example (sub02/EP03_02f on the CASME II) of the sparse attention matrix $\sum_{\left(i,j\right)\in \Omega_{i,j}}\mathrm{Softmax}\left(\frac{q_{i}k_{j}^{T}}{\sqrt{d_{k}}}\right)$ where darker color denotes larger values. The rows represent the index of q and the columns represent the index of k.

**Table 1 entropy-24-01271-t001:** Hyper-parameters of the UAMRN.

Hyper-Parameter	Description	Value
$n_{p}$	Number of hyper-planes	3
*L*	Length of the rebuilt sequence	32
$n_{t}$	Number of hash tables	4
$n_{h}$	Number of attention heads	2
AFR	Amplification factor range	[2:1:13]
$d_{model}$	Dimension of the model	1024

**Table 2 entropy-24-01271-t002:** Experimental Results (Accuracy/F1-score) on the CASME II with 5 classes. The bold font indicates best performance.

Methods	Accuracy (%)	F1-Score (%)
LBP-TOP + AdaBoost (2014) [42]	43.78	33.37
STRBP (2017) [43]	64.37	N/A
HIGO + Mag (2018) [26]	67.21	N/A
ME-Booster (2019) [23]	70.85	N/A
TSCNN (2019) [44]	**80.97**	**80.70**
Graph-tcn (2020) [25]	73.98	72.46
AU-GCN (2021) [29]	74.27	70.47
**AUMRN**	80.57	76.02

**Table 3 entropy-24-01271-t003:** Experimental Results (Accuracy/F1-score) on the SAMM with 5 classes. The bold font indicates best performance.

Methods	Accuracy (%)	F1-Score (%)
DSSN (2019) [45]	57.35	46.44
TSCNN (2019) [44]	71.76	69.42
Graph-tcn (2020) [25]	75.00	69.85
MTMNet (2020) [46]	74.10	73.60
AU-GCN (2021) [29]	74.26	70.45
GEME (2021) [47]	65.44	54.67
MERSiamC3D (2021) [20]	64.03	60.00
**AUMRN**	**79.41**	**77.32**

**Table 4 entropy-24-01271-t004:** Experimental Results (Accuracy/F1-score) on the SMIC-HS with 3 classes. The bold font indicates best performance.

Methods	Accuracy (%)	F1-Score (%)
LBP-SIP (2014) [48]	62.80	N/A
STRBP (2017) [43]	60.98	N/A
Dual-Inception Network (2019) [49]	66.00	67.00
3D-CNNs (2019) [21]	66.30	N/A
TSCNN (2019) [44]	72.74	72.36
MTMNet (2020) [46]	76.80	74.40
GEME (2021) [47]	64.63	61.58
**AUMRN**	**78.05**	**78.14**

**Table 5 entropy-24-01271-t005:** Effect of Uncertainty Estimation. The bold font indicates best performance.

	Acc (%)			F1-Score (%)		
	CASME II	SAMM	SMIC-HS	CASME II	SAMM	SMIC-HS
Deterministic	78.95	76.47	77.44	74.17	74.51	76.75
Uncertainty-Aware	**80.57**	**79.41**	**78.05**	**76.02**	**77.32**	**78.14**

**Table 6 entropy-24-01271-t006:** Effect of Sparsity Constraints. The bold font indicates best performance.

	Time Complexity O (N)	Acc (%)		
		CASME II	SAMM	SMIC-HS
Full-Attention	$bn_{h}L^{2}$	78.54	77.21	74.39
SSA	$bn_{h}n_{t}w{\left(\frac{L}{2w}\right)}^{2}$	**80.57**	**79.41**	**78.05**

## Data Availability

Not applicable.

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
