# Peer review of "Micro-Expression Recognition Using Uncertainty-Aware Magnification-Robust Networks"

_entropy, 2022, doi:10.3390/e24091271_

Round 1

Reviewer 1 Report

This paper proposes a magnification-robust network to tackle the magnification noise problem. The work is well-motivated, the solution sounds reasonable and the experiments seem convincing. Here are some comments:

(1) Why did you choose the amplification factor range as [2 : 1 : 13]? 

(2) "Some ME samples may be labeled incorrectly since the stimulus materials to elicit ME may differ from the actual emotion displayed by the participant." This sentence is unclear, and please reclarify it for better understanding.

(3) What is the number of training iterations in your experiments?

(4) Why did you use the apex and onset frame to produce magnified images?

Reviewer 2 Report

This paper is well organized. It provides a new insight to encode ME movements and tackle magnification noise. 

(1) This paper should provide the comparison result on other state of the art methods as it mentioned in the abstract: "The experiments on three public MER databases demonstrate our superiority against the state-of-the-art methods"

(2) Does the word "artifacts" in the paper means noise? How is it reflected in the image?

(3) Why the number of sparse self-attention (SSA) blocks is three?

(4) What is the aleatoric and epistemic in the uncertainty estimation?
